# Tracing G-Protein-Mediated Contraction and Relaxation in Vascular Smooth Muscle Cell Spheroids

**DOI:** 10.3390/cells12010128

**Published:** 2022-12-28

**Authors:** Jaspal Garg, Alexandra Sporkova, Markus Hecker, Thomas Korff

**Affiliations:** 1Institute of Physiology and Pathophysiology, Department of Cardiovascular Physiology, Heidelberg University, 69120 Heidelberg, Germany; 2European Center for Angioscience (ECAS), Medical Faculty Mannheim, Heidelberg University, 69120 Heidelberg, Germany

**Keywords:** G-protein signaling, RGS, VSMC phenotype, MAPK, contraction, relaxation

## Abstract

Analyses of G-protein-mediated contraction and relaxation of vascular smooth muscle cells (VSMCs) are usually hampered by a rigid growth surface and culture conditions promoting cell proliferation and a less contractile phenotype. Our studies indicated that mouse aortic VSMCs cultured in three-dimensional spheroids acquire a quiescent contractile status while decreasing the baseline G-protein-dependent inositolphosphate formation and increasing the expression of endothelin receptor type A (*Ednra*). Endothelin-1 (ET-1) promoted inositolphosphate formation in VSMC spheroids, but not in VSMCs cultured under standard conditions. To trace ET-1-mediated contraction of VSMC spheroids, we developed an assay by adhering them to collagen hydrogels and recording structural changes by time-lapse microscopy. Under these conditions, mouse and human VSMC spheroids contracted upon treatment with ET-1 and potassium chloride or relaxed in response to caffeine and the prostacyclin analogue Iloprost. ET-1 activated AKT-, MKK1-, and MKK3/6-dependent signaling cascades, which were inhibited by an overexpressing regulator of G-protein signaling 5 (*Rgs5*) to terminate the activity of Gα subunits. In summary, culture of VSMCs in three-dimensional spheroids lowers baseline G-protein activity and enables analyses of both contraction and relaxation of mouse and human VSMCs. This model serves as a simple and versatile tool for drug testing and investigating G-protein-depending signaling.

## 1. Introduction

Trimeric G proteins are composed of α, β and γ subunits, which constitute the top hierarchy of determinants controlling contraction and relaxation of vascular smooth muscle cells (VSMCs) [1]. Upon binding of stimulatory agonists to G-protein coupled receptors (GPCRs), VSMC contraction or relaxation is regulated via one or several types of Gα subunits, including Gα_q/11,_ Gα_12/13_, Gα_i/o_ and Gα_s_. Upon stimulation Gα subunits replace guanosine diphosphate (GDP) by guanosine triphosphate (GTP) and activate specific target effector proteins [2]. This process is accompanied by the dissociation of βγ-subunits, which are capable of stimulating further signaling cascades comprising multiple kinases, such as mitogen-activated protein kinases (MAPKs) [3]. As can be deduced from the complexity of this signal transduction mechanism and considering that several types of α-subunits may be activated by one agonist, contractile responses of VSMCs are regulated via multiple input pathways. As such, endothelin-1 (ET-1) may stimulate several signaling cascades in aortic VSMCs at once upon binding to type-A ET-1 receptors (ET_A_). This promotes the intracellular release of calcium via the Gα_q/11_-PLC-IP-3 pathway, activation of Rho-associated kinase via Gα_12/13_-RhoGEF [4,5], stimulation of protein kinase A (PKA) via Gα_i_-coupled βγ-subunits [6], and activation of several MAPKs [5]. On the functional level, these signaling events converge to promote a robust and long-lasting constriction of arterial VSMCs.

The outcome of G-protein-mediated signaling events critically depends on the expression of GPCRs, which is altered by environmental conditions promoting the dedifferentiation of VSMCs [7]. As such, cell culture conditions optimized to feature the growth of VSMCs while suppressing a contractile phenotype challenge analyses of the G-protein-dependent signaling events and the assessment of the therapeutic capacity of pharmacological agents. Moreover, the high rigidity of culture surfaces limit or even prohibit the observation of regular contractile responses to G-protein-activating agonists. Especially VSMC relaxation is difficult to assess as tensile stress originating from the environment is required to reshape the VSMC morphology upon a decrease in the tension generated by the contractile apparatus. Several experimental setups have been developed to overcome these analytical limitations, including the determination of the percentage of VSMCs changing their morphological shape in response to a stimulus [4]. By growing VSMCs on artificial silicone-based substrates, the formation of distortions of the extracellular matrix has been utilized as indirect readout to determine the increase or decrease in VSMC tension [8]. In a more recent study, elegant magnetic three-dimensional (3D) bioprinting-based techniques were applied to generate vasoactive ring-like VSMC structures for drug testing purposes [9]. Assessment of contractile responses is meanwhile supported by modern computational techniques coping with the phenotype heterogeneity of cultured VSMCs promoted by the cell culture environment [10].

While all these models allow for the analyses of contractile responses, most of them either rely on proliferating VSMCs cultured as monolayer, require long-term culture of VSMCs or sophisticated technical equipment. In this study, we cultured VSMCs in three-dimensional spheroids, which were shown to promote cellular quiescence without applying starvation conditions and maintain their contractile phenotype [11]. They responded to G-protein-activating agonists, such as angiotensin II and norepinephrine, by increasing intracellular calcium levels while allowing for analyses of downstream signaling events [12]. VSMC spheroids are generated from suspended VSMCs within 24 h in hanging drops or in non-adhesive U-shaped 96-well plates if specific treatment options for subsets of spheroids are required. Here, we tested the general suitability of VSMC spheroids as a tool for the analysis and modification of G-protein-mediated responses on the level of signaling and function. In this context, we intended to develop a simple assay allowing the quantification of both VSMC contraction and relaxation without sophisticated equipment or specialized analytical tools.

## 2. Materials and Methods

### 2.1. Antibodies and Reagents

The rabbit anti-human/mouse Endothelin Receptor A polyclonal and rabbit anti-human/mouse Valosin-Containing Protein (VCP) antibody was purchased from Invitrogen (Carlsbad, CA, USA, PA3-065) and Novus Biologicals (Centennial, CO, USA. NB100-1558), respectively. G_q/11_ protein inhibitor YM-25490 was purchased from Fujifilm Wako Chemicals (Neuss, Germany, 257-00631). Endothelin-1 (1160) and Iloprost (2038) from Tocris (Bristol, UK) and Caffeine from Sigma-Aldrich (Waltham, MA, USA, C0750). All other used chemicals were of analytical grade.

### 2.2. Cell Culture

Murine aortic SMCs (aoSMC) were isolated from the aorta of 6–8 weeks old *Rgs5^fl/fl^* mice, as previously reported [12,13]. In brief, after carefully removing the adventitia, the aorta was washed twice in Dulbecco’s PBS (without calcium and magnesium). It was cut in 1-mm–sized rings and digested overnight with 1% collagenase (C5138, Merck GmbH, Darmstadt, Germany). aoSMC were resuspended, seeded on culture plates, and routinely checked for expression of SMC marker proteins (e.g., αSMA, SMMHC) and utilized until passage 5 for experiments. Human umbilical arterial smooth muscle cells (HUASMCs) were purchased from ProVitro ( Berlin, Germany, 1110611) and cultured up to passage 8 for all experiments. aoSMC and HUASMC were cultured in low-glucose DMEM (Thermo Fisher Scientific, Waltham, MA, USA) containing 15% fetal calf serum (Pelobiotech, Planegg, Germany).

### 2.3. Adenoviral Transfection

Murine SMCs (aoSMCs) isolated from *Rgs5^fl/fl^* mice were transduced with adenoviruses (MOI 1000) carrying empty vector (Ad-CMV-Null, Vector Biolabs, 1300, Malvern, PA, USA) or *Rgs5* (Ad-CMV-Rgs5, Vector Biolabs, 270494, Malvern, PA, USA) to overexpress *Rgs5*. mRNA analysis was performed 72 h after transduction by qPCR to verify the overexpression of *Rgs5*.

### 2.4. Generation of 3D Spheroids

Murine aoSMC or HUASMCs were detached with trypsin, centrifuged at 1000 rpm for 5 min and counted in a Neubauer chamber or an automatic cell counter (CASY, OLS, Germany). Droplets of 25 µL culture medium with 0.24% *w*/*v* methyl cellulose (M0650, Sigma-Aldrich/Merck, Darmstadt, Germany) and 15% FCS containing 3000 (HUASMC) or 500 (aoSMC) cells were pipetted onto squared petri dishes, which were then turned bottom up for generating hanging drops. VSMCs aggregate to form 3D spheroids within 24 h. They were harvested by flushing the plate with 10 mL of DPBS per dish (14040091, Thermo-Fisher), centrifuged at 1000 rpm for 5 min, and processed for further application.

### 2.5. Collagen Hydrogel Preparation

Collagen stock solution (4.5 mL, ~2 mg/mL type I collagen prepared from rat tails in 0.1% acetic acid [14]) was carefully mixed with 500 μL of 10X M199 (Sigma-Aldrich, M0605) and quickly neutralized by adding approx. 375 μL of sterile 0.2 M NaOH. Subsequently, the same volume DMEM supplemented with 30% FCS was added. Hydrogels were prepared by rapidly distributing 0.5 mL of the final collagen hydrogel mixture in each well of a pre-warmed 24-well plate, which was incubated for 30 min at 37 °C to support polymerization of the hydrogels. Homogeneous polymerization and collagen concentration are critical determinants of hydrogel rigidity and spheroid attachment.

### 2.6. Contraction/Relaxation Assay

Approximately 10 spheroids in 0.5 mL growth medium were homogeneously distributed on the surface of each hydrogel. Hydrogels were incubated at 37 °C/5% CO_2_ in a humidified chamber for 18–24 h to allow attachment of the VSMC spheroids. Five to eight spheroids with no other spheroids located in their perimeter (>500 µm) were selected for observation per treatment group. Structural changes of attached spheroids were imaged by combining time-lapse (interval: 10 s for up to 7 min) and phase–contrast microscopy (Olympus IX3, Hamamatsu C10600 camera, Tokyo, Japan). Agonists (or solvent as control) were applied in 0.5 mL pre-warmed DMEM supplemented with 15% FCS (DMEM/FCS). Automated frame-based structural segmentation analyses of VSMC spheroids were performed by utilizing the cellSens^®^ software (Olympus, Tokyo, Japan, version 1.18). Area changes were assessed and quantified, as shown in Appendix A.

### 2.7. IP-1 Assay

Intracellular concentrations of IP-1 were quantified in aoSMC, 3D spheroids or murine aortas using the HTRF-IP1 kit (62IPAPEB, Cisbio, Codolet, France) following the manufacturer’s instructions. Briefly, aoSMC were trypsinized, cell pellet was washed with PBS, and then resuspended in stimulation buffer containing 50 mM LiCl. aoSMC and transferred to a 384-well microtiter plate at a density of 10,000 cells per well in 7 µL of stimulation buffer. After 10 min of incubation, 7 µL of stimulation buffer containing indicated substances (100 nM of ET-1 or 10 μM of YM-254890) was added to the wells for 1 h. 3 μL of IP-1-d2 conjugate followed by 3 µL of europium cryptate-labelled anti-IP-1 antibody both dissolved in lysis buffer were then added to the wells. After a further dark incubation of 1 h at room temperature, time-resolved fluorescence was measured at 620 and 665 nm with the PheraStar (BMG Labtech, Ortenberg, Germany) multimode plate-reader. Using a standard curve generated from the IP-1 standard solution provided by the manufacturer, HTRF ratios were converted to IP-1 concentrations in nM for unknown samples.

For measurement in aortas and 3D spheroids, aortas/spheroids were stimulated in stimulation buffer with YM-254890 or ET-1 for 1 h. After centrifugation, they were lysed in lysis buffer provide by the manufacturer and 7 µL of these lysates were used for the measurement. Total protein content in the lysates was determined to normalize IP-1 levels.

### 2.8. cAMP Assay

Intracellular concentrations of the second messenger cAMP were quantified in aoSMC or 3D spheroids using the HTRF-cAMP-Gi kit (62AM9PEB, Cisbio, Codolet, France) following the manufacturer’s instructions. Briefly, aoSMC were trypsinized, cell pellet was washed with PBS, and then resuspended in stimulation buffer provided by the manufacturer. aoSMC were transferred to a 384-well microtiter plate at a density of 5000 cells per well in 5 µL of stimulation buffer. After 10 min of incubation, 5 µL of stimulation buffer containing 100 nM ET-1 was added for 1 h. 5 μL cAMP d2-labeled antibody followed by 5 µL of europium cryptate-labelled cAMP both dissolved in lysis buffer were added to the wells. After a further dark incubation of 1 h at room temperature, time-resolved fluorescence was measured at 620 and 665 nm with the PheraStar (BMG LabTech) multimode plate reader. A standard curve was generated with the cAMP standard solution provided by the manufacturer, and all HTRF ratios were converted to cAMP concentrations in nM for unknown samples. For measurement in 3D spheroids, spheroids were stimulated in stimulation buffer containing 100 nM ET-1 for 1 h. After centrifugation, they were lysed in lysis buffer provide by the manufacturer, and 5 µL of these lysates were used for the measurement. Total protein content in the lysates was determined to normalize cAMP levels.

### 2.9. Quantitative Real Time RT-PCR (qPCR) Analysis

RLT Buffer (79216, Qiagen) containing 1% β-Mercaptoethanol was used to lyse aoSMC. Total RNA was isolated using the RNeasy Mini Kit (74601, Qiagen, Hilden, Germany) according to the manufacturer’s instructions. cDNA was synthesized using the Omniscript Reverse Transcription Kit (205113, Qiagen) and quantitative real-time PCR for the target sequences was performed at the Rotor-Gene Q (Qiagen) device using the EvaGreen Master Mix (BS76.590.5000, Bio & SELL, Feucht, Germany). Amplification of the 60S ribosomal protein L32 (RPL32) cDNA served as an internal reference. Fluorescence was monitored (excitation at 470 nm and emission at 530 nm) at the end of the annealing phase (60 °C), and the threshold cycle (Ct) was set within the exponential phase of the PCR. Using the ΔΔCt method, quantification of the PCR product was performed. The following primer pairs were used: mRgs5 5′ GCGGAGAAGGCAAAGCAA 3′, 5′ GTGGTCAATGTTCACCTCTTTAGG 3′; mRpl32 5′ GGGAGCAACAAGAAAACCAA 3′, 5′ ATTGTGGACCAGGAACTTGC 3′; mEdnra 5′-GCTGGTTCCCTCTTCACTTAAGC-3′, 5′-TCATGGTTGCCAGGTTAATGC -3

### 2.10. Analysis of Kinase Phosphorylation

To simultaneously determine the relative phosphorylation level of multiple kinases, a MAPK Pathway phosphorylation Array Kit (RayBiotech, Cologne, Germany, AAH-MAPK) was applied according to manufacturer’s instructions. Briefly, 3D aoSMC lysates were incubated with the membranes, where (phospho)kinase-specific primary antibodies have been spotted in a predefined manner. After washing, membranes were incubated with biotinylated detection antibodies. Streptavidin-HRP was added for chemiluminescent detection, which was performed with Image Quant™ TM LAS 4000 MINI (GE Healthcare). Grey intensities were quantified using the Image Quant TL Software (GE Healthcare, Waukesha, WI, USA, Version 8.1).

### 2.11. Statistical Analysis

All results are expressed as means ± SD as indicated and statistically analyzed by utilizing GraphPad Prism (Version 9.1). Outliers were identified by application of the Grubbs’ test with α set to 0.05. Differences among normally distributed values of two individual experimental groups were analyzed by unpaired Student’s t-test. Differences of one parameter between normally distributed values (Shapiro-Wilk test) of three experimental columns were analyzed by one-way ANOVA followed by Šídák’s multiple comparisons test for selected pairs of columns. Grouped data were analyzed using two-way ANOVA followed by Šídák’s multiple comparisons test to compare selected pair of columns. *p* < 0.05 was considered statistically significant (* *p* < 0.05, ** *p* < 0.01, *** *p* < 0.001).

## 3. Results

### 3.1. Three-Dimensional Culture of VSMCs in Spheroids Increases Sensitivity to ET-1

Standard two-dimensional (2D) culture conditions promote an activated phenotype and stimulate proliferation of VSMCs, which may affect their responsiveness to humoral stimuli. We observed that mouse aortic VSMCs lose their sensitivity to ET-1 upon exposure to standard 2D culture conditions (Figure 1A), as shown by analyzing generation of IP-1—a metabolite whose level correlates with the production of IP-3 by phospholipase C β (PLCβ). Although ET-1 stimulation increased IP-1 generation in mouse aortas (Figure 1B), no such response was observed in cultured VSMCs (Figure 1C). Similarly, ET-1 stimulation did not alter cAMP generation in these cells (Appendix A). Moreover, treatment of VSMCs exposed to control culture conditions with the Gα_q/11_ inhibitor YM-254890 induced a robust decline in IP-1 production (Figure 1D), indicating a significant baseline activity of the Gα_q/11_-PLCβ signaling axis. In contrast, analyses of baseline IP-1 generation in VSMCs cultured in 3D spheroids (Figure 1E) indicated a low Gα_q/11_-PLCβ activity level (Figure 1F), which was significantly elevated by stimulation with ET-1 (Figure 1G). As shown for 2D culture conditions, cAMP levels were not altered in VSMC spheroids upon ET-1 stimulation (Appendix A). Gene expression studies of human VSMCs suggested that 3D culture conditions promote the expression of the endothelin receptor type A (*Ednra*) but not type B (*Ednrb)* that was partially dependent on TGFβ1 signaling (Appendix A) [11]. Likewise, *Ednra* expression was increased in 3D spheroids generated from mouse aortic VSMCs as compared to 2D culture conditions (Figure 1H).

### 3.2. Recording and Quantification of Contraction and Relaxation of VSMC Spheroids Adhered to Collagen Type I Matrices

Based on the initial observations, we next investigated the functional outcome of the ET-1 stimulation. To this end, we exploited the features of VSMC spheroids and developed an assay for recording their contractile responses. Single spheroids were adhered to the surface of a collagen type I hydrogel. The rigidity of the gel was optimized to allow for cell spreading on the one hand and for retracting the matrix on the other hand. Initial attachment and spreading of VSMCs required 18–24 h, after which the gels were transferred to a phase contrast microscope. Morphological changes were traced by time-lapse recording upon stimulation with different agonists. The general usability of the model was tested by treating the spheroids with potassium chloride to induce VSMC depolarization and subsequent contraction triggered by an increase in intracellular calcium levels via activation of L-type calcium channels. In addition, we tested the response to caffeine, which predominantly supports relaxation by non-selective inhibition of phosphodiesterases and preservation of cAMP levels in VSMCs [15]. Quantification of the morphological changes were conducted by automated segmentation analyses to identify cellular structures, followed by overlying images taken before (red colored segmentation) and at the end (green colored segmentation) of the constriction and relaxation, respectively. Afterwards changes in the structural area were assessed (Appendix A). All functional responses to the stimuli occurred within 5 min and were recorded by utilizing time-lapse microscopy (Exemplary videos are available as Appendix A). Control treatment did not alter the morphology of adhered spheroids as evidenced by the unchanged position of the structural cellular elements (Figure 2A,D, overlapping structures in yellow). Stimulation with potassium chloride induced a contraction (Figure 2B,D) as evidenced by the decrease in the distance of individual cellular structures when comparing their position before and after application of the stimulus (Figure 2B). The structures ‘move’ towards an area of the spheroid that appeared to serve as ‘anchor point’ (Figure 2, encircled area) with no or minimal positional change in the structural elements (overlapping structures in yellow). Caffeine relaxed the spheroid structure as evidenced by the increase in the distance of individual cellular structures moving away from the anchor point (Figure 2C,D).

### 3.3. ET-1-Mediated VSMC Contraction Is Attenuated by RGS5

We next overexpressed RGS5 (Appendix A) to test whether interference with Gα_q/11_- and Gα_i/o_-dependent signaling alters the contractile response evoked by the treatment with ET-1. While RGS5 had no impact on baseline *Ednra* expression (Appendix A), its effect on ET-1-induced G-protein signaling in VSMC spheroids was assessed by IP-1 generation revealing a partial but not complete suppression of Gα_q/11_-PLCβ activity (Figure 3A). Likewise, RGS5 only partially inhibited ET-1 induced VSMC contraction (Figure 3B). Correspondingly, ET-1 evoked moving distance of structural elements was slightly reduced but not completely inhibited by RGS5 (Figure 3C, overlapping segments in yellow). We additionally analyzed as to how ET-1 and RGS5 alter downstream signaling cascades by assessing the level of specific phosphorylation sites known to activate protein kinases and their substrates. ET-1 stimulated the phosphorylation of AKT and its target P70S6K (Figure 4A), both of which were attenuated by RGS5 (Figure 4B). Analyses of the MKK1/ERK1-dependent signaling pathway and the ERK1/2 target RSK1 showed similar results, but no RGS5-dependent inhibition of ERK1/2 phosphorylation at the investigated time point (Figure 4C,D). MKK3/6-dependent signaling was stimulated by ET-1 and inhibited by RGS5 (Figure 4E,F). However, while phosphorylation of the p38 targets HSP27 and MSK2 were significantly altered, phosphorylation of p38 appeared quite heterogeneous (Figure 4E,F).

### 3.4. Functional Analysis of G-Protein-Mediated Responses of Human VSMCs by the Spheroid-Based Contraction/Relaxation Assay

The results obtained so far showed that the spheroid-based contraction/relaxation assay is suitable for the assessment of G-protein-dependent signaling and functional responses of murine VSMCs. Finally, we tested whether human VSMCs respond in a comparable fashion. Corresponding experiments with human umbilical artery smooth muscle cells (HUASMCs) indicated that ET-1 treatment promoted VSMC contraction as compared to DMSO (Figure 5A). We next determined the impact of the prostacyclin analogue Iloprost on the tone of human VSMCs and observed that this drug reproducibly induced a sustained relaxation. The dynamics of corresponding morphological changes were exemplarily assessed by an automated frame-based analysis encompassing the structural alteration of detected cellular segments. While both ET-1 and Iloprost stimulation induced a rapid directed response indicating VSMC contraction or relaxation, respectively, DMSO (solvent control) treatment evoked no such directed responses.

## 4. Discussion

In vitro culture of VSMCs is usually achieved by growing primary arterial smooth muscle cells on plastic surfaces as two-dimensional monolayers while being exposed to medium formulations supplemented with growth factors and fetal calf serum. These conditions are fundamentally different from their natural environment and support proliferation while decreasing the degree of differentiation, as evidenced by the successive loss of markers defining their contractile phenotype, such as alpha smooth muscle actin (αSMA) and smooth muscle myosin heavy chain (SMMHC) [16]. These proteins constitute structural elements of the contractile apparatus that in vivo generates the baseline tone of VSMCs and thus regulates the diameter of blood vessels. Consequently, the contractile capacity of VSMCs may be limited under standard cell culture conditions. Moreover, cultured VSMCs may also change the expression pattern of GPCRs, regulating the baseline tone, as was shown for rat aortic smooth muscle cells, which invert the ET_A_/ET_B_ receptor ratio while being propagated in vitro [17].

In light of these findings, we investigated whether VSMCs cultured in three-dimensional spheroids may serve as a suitable tool for tracing functional responses to G-protein-mediated stimuli. Earlier studies indicated that the differentiation of VSMCs is altered by organizing them in three-dimensional spheroids. In these aggregates, proliferation of VSMCs ceases, which is partially mediated by the autocrine release of transforming growth factor beta (TGFβ) [11]—an important regulator of VSMC differentiation [18,19]. Moreover, gene expression associated with protein biosynthesis is downregulated while the level of proteins of the contractile apparatus is elevated or maintained. Collectively, these observations indicated that culturing of VSMCs in spheroids support their quiescent differentiated phenotype, which is why these were initially described as ‘in vitro analogue of the arterial wall’ [20].

Here, we show that VSMCs in three-dimensional spheroids regain their sensitivity to ET-1 at least with respect to the Gα_q/11_-PLC-IP-3 signaling axis. Increased expression of the ET_A_ receptor serves as the most likely explanation for this finding, which may partially be controlled by TGFβ considering the significant drop in ET_A_ receptor expression after inhibition of the corresponding signaling pathway in human VSMCs (Appendix A, [11]). Interestingly, a significant Gα_q/11_-PLC-IP-3 baseline activity was detected in VSMCs under standard culture conditions, as was also shown for several types of Gα subunits in cultured bovine aortic smooth muscle cells [4]. This agonist-independent G-protein activity was fully silenced in three-dimensional spheroids possibly caused by the elevated level of RGS5 detected in VSMC spheroids [12] that is able to terminate Gα_q/11_ activity by its GTPase activating properties. On the contrary, loss of RGS5 in VSMC spheroids was associated with an elevated intracellular calcium level as the result of unrestricted Gα_q/11_-PLCβ-IP-3 activity [12]. As such, RGS5 appears to be relevant for the control of baseline G-protein activity in general.

Despite being tempered by endogenous RGS5, ET-1 stimulated Gα_q/11_-PLCβ-IP-3 activity in VSMC spheroids. Given that this stimulus did not affect the cAMP levels, triggering of Gα_i/o_-dependent signaling appears unlikely in this context. Consequently, the observed ET-1-induced modification of downstream kinase activity is primarily mediated by the activation of Gα_q/11_- and corresponding βγ-subunits, both of which may directly or indirectly promote phosphorylation and activation of AKT and MAPKs [21,22,23]. Moreover, binding of agonists to GPCRs also supports the formation of scaffolds based on β-arrestins to support the G-protein independent interaction with and activation of other determinants of intracellular signaling cascades, including MAP kinases, PI3 kinase (PI3K), and AKT [24]. However, as the phosphorylation of most of these kinases was attenuated upon overexpression of RGS5, Gα_q/11_-dependent signaling appears to be the most likely source of MAPK and PI3K/AKT stimulation. Additionally, ET-1 triggers the activation of Gα_12/13_-Rho signaling, which was not specifically addressed in this study. The relevance of this pathway for the contraction of VSMCs may nevertheless be indirectly deduced from the fact that RGS5 only partially blocked the contractile response to ET-1. The Gα_12/13_-Rho signaling axis may, in fact, be amplified by RGS5, which was shown to promote RhoA activity in vitro and in vivo [12,13].

The diversity of signaling cascades triggered upon ET-1 stimulation underlines the relevance of assessing their functional impact on contractile responses of VSMCs. Especially in view of functional testing of pharmacological compounds, the development of corresponding assays is challenging as culture conditions usually promote cellular proliferation and a less contractile phenotype and may significantly diminish functional responses to distinct agonists as shown for ET-1. To overcome such limitations, we intended to develop a simple assay that supports both a stable quiescent contractile VSMC phenotype and assessment of both contraction and relaxation without the need of specialized equipment. While contraction-mediated shape changes of spheroids may in principle be directly traceable, their relaxation requires an intrinsic physical force sufficient to expand cells without interfering with their contraction. For that reason, we decided to attach the VSMC spheroids to collagen type I hydrogels, which were optimized to support cell attachment and ‘store‘ tensional forces generated by the overall tone of the focally accumulated VSMCs. In fact, forces generated by cellular traction are known to modify the mechanical properties of extracellular matrices [25]. Those tensional forces originating from VSMCs were impressively visualized by the formation of wrinkles and distortions on polymerized silicon surfaces [8]. Similarly, attachment of spheroid-derived endothelial cells was shown to generate tensional forces, which modify the structure of collagen hydrogels [26]. The mechanical interaction of cells and the extracellular matrix is elegantly described by the tensegrity model [27]. Cell-derived traction is generated by intracellular contractile elements and transferred to the matrix via focal adhesions. Consequently, any change in the force generated by the contractile apparatus of VSMCs is immediately transmitted to the matrix. Our model exploits this mechanism to visualize VSMC contraction or relaxation. Potassium chloride depolarizes VSMCs and triggers a calcium-calmodulin-dependent activation of the myosin light chain (MLC) kinase that phosphorylates the MLC to promote contraction [28]. ET-1-induced activation of the ET_A_ receptor comprise a more complex interaction of the Gα_12/13_- and Gα_q/11_-dependent pathways promoting the phosphorylation of MLC by Rho-, calcium- as well as PKC-dependent signaling mechanisms. Relaxation of VSMC spheroids were also traceable, as shown for caffeine. Although this plant-derived alkaloid may in principle evoke a fast release of calcium from intracellular stores supporting a transient contraction in certain vascular beds [29], relaxation appears to be the preferential response of VSMCs [15]. Mechanistically, this may partially be based on the caffeine-mediated competitive inhibition of phosphodiesterases preserving the cAMP level as a prerequisite for protein kinase A (PKA)-regulated MLC dephosphorylation. Similarly, the prostacyclin analogue Iloprost binds to the prostacyclin receptor (IP1)—a GPCR that activates the Gα_s_-cAMP-PKA signaling pathway to promote MLC dephosphorylation and thus VSMC relaxation [30].

All aforementioned stimuli rapidly contracted or relaxed VSMCs as predicted by the literature and were traced intentionally by applying simple analytical techniques. We combined time-lapse recording and computer-assisted segmentation analyses to identify and quantify the corresponding morphological changes of VSMC spheroids. However, the analysis of their overall response may also be simplified by processing (e.g., by using open-source software ImageJ) individual phase contrast images taken before and some minutes after applying the stimulus when the response maximum has been reached. On the other hand, application of more sophisticated morphometric software tools may increase sensitivity and resolution of time-related analyses. With respect to increasing the throughput, hydrogels with a dished meniscus may also be generated in 96-well plates, which allow placing of single spheroids in the center of each gel for standardized recording protocols. Finally, endothelial cells (ECs) may be added to VSMC spheroids during their formation to generate vascular organoids composed of a superficial endothelial monolayer covering a core of VSMCs [31,32]. Such a model would broaden the options for analyses and include the endothelium as another important element controlling the tone of VSMCs.

## 5. Conclusions

Collectively, culture in three-dimensional spheroids supports a quiescent contractile phenotype of VSMCs, which show a low baseline Gα_q/11_ activity and are sensitive to GPCR-mediated stimuli. Upon attaching the spheroids to hydrogels, agonist-induced contraction or relaxation of VSMCs can easily be traced and quantified. The level of these responses may be modified by interfering with the activation of G-protein-dependent signaling cascades as evidenced by artificially increasing RGS5 levels. Finally, this assay can be applied to mouse and human VSMCs and may serve as a simple tool for the functional screening of pharmacological compounds.

## Figures and Tables

**Figure 1 cells-12-00128-f001:**
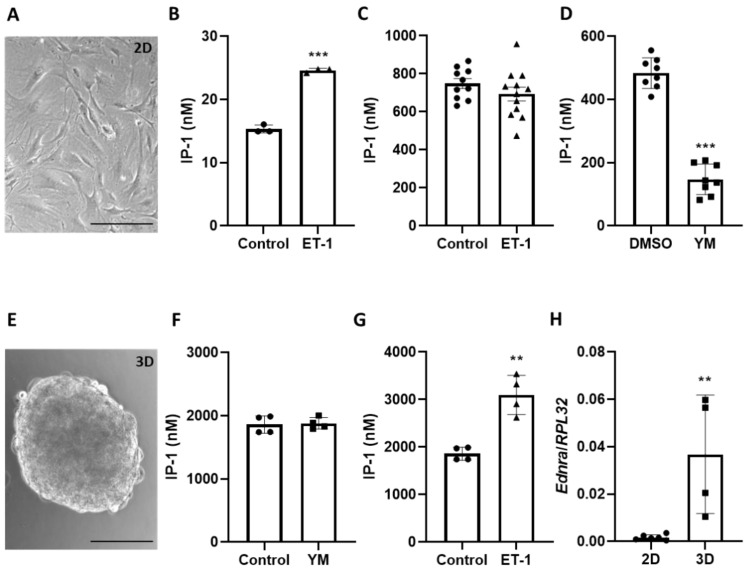
Comparison of ET-1-dependent Gα_q_-protein signaling in 2D and 3D VSMC. (**A**) Representative phase-contrast image (20-fold magnification) of murine aoSMC cultured as a 2D monolayer (scale bar: 100 µm). (**B**–**D**) Analyses of the IP-1 level in ET-1-treated (100 nM) murine aortas ((**B**), n = 3, *** *p* < 0.001) and in aoSMC cultured as monolayer upon ET-1 treatment ((**C**), n = 10–12, data were normalized to total protein levels) or treated with the Gα_q/11_ inhibitor YM-254890 (10 µM) ((**D**), n = 8, *** *p* < 0.001). (**E**) Representative phase-contrast image (20-fold magnification) of aoSMC cultured as 3D spheroids (scale bar: 100 µm). (**F**,**G**) IP-1 level measured in 3D aoSMC treated with YM-254890 ((**F**), n = 4) or ET-1, ((**G**), n = 4, ** *p* < 0.01). (**H**) *Ednra* expression was determined by qPCR in RNA samples isolated from 2D or 3D cultured murine aoSMC (n = 4–5, ** *p* < 0.01, *RPL32* was used as reference).

**Figure 2 cells-12-00128-f002:**
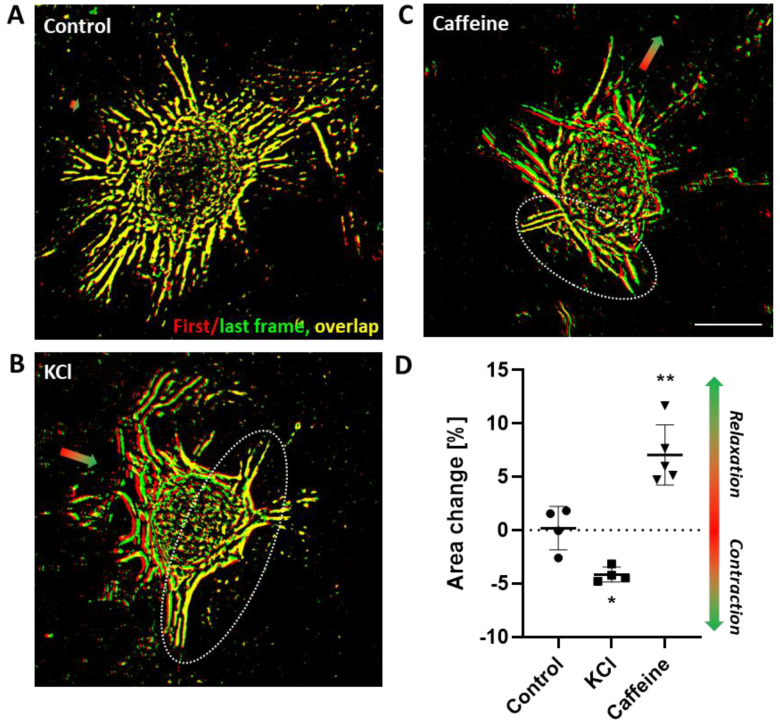
In vitro contraction/relaxation assay. (**A**–**C**) Representative images showing structural segments (computer-assisted detection) of aoSMC spheroids attached to collagen type I hydrogels before (red) and after (green) being stimulated with solvent ((**A**), control), potassium chloride ((**B**), KCl, 60 mM) or caffeine ((**C**), 10 mM, scale bar: 100 µm). The dotted circle defines the area that serves as ‘anchor’ for the contracting/relaxing spheroid. Colored arrows indicate the accumulated movement vector of the cellular segments in relation to the anchor point. (**D**) Quantification of area changes (see Appendix A) of aoSMC spheroids upon exposure to the stimuli (n = 4–5, * *p* < 0.05, ** *p* < 0.01).

**Figure 3 cells-12-00128-f003:**
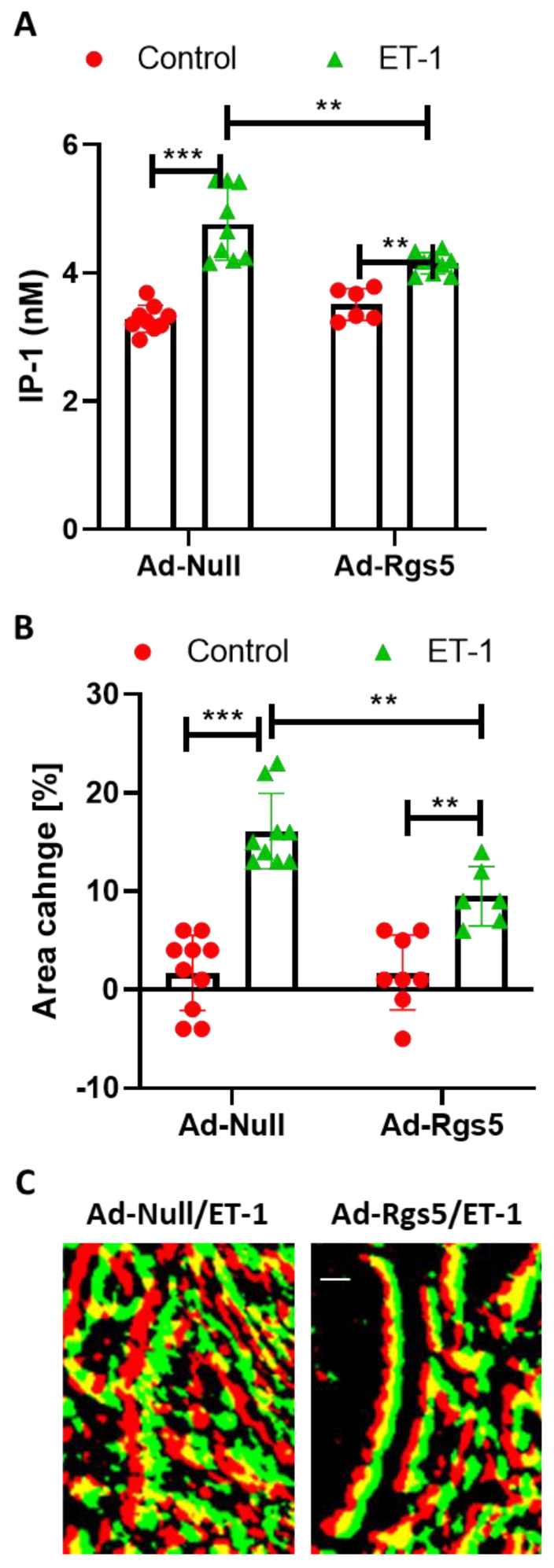
Inhibition of ET-1-induced contractile responses by RGS5. Murine aoSMC were transduced with adenoviral vectors carrying either empty vector (Ad-Null) or *Rgs5* (Ad-Rgs5). (**A**) The IP-1 level was measured in three-dimensional spheroids treated with solvent (control) or ET-1 (100 nM, n = 6–9, ** *p* < 0.01, *** *p* < 0.001, statistical comparisons as indicated). (**B**) Quantification of area changes of Ad-Null- or Ad-Rgs5-transduced aoSMCs induced by ET-1 (n = 6–10, ** *p* < 0.01, *** *p* < 0.001, statistical comparisons as indicated). (**C**) Representative (enlarged) overlay of segments of in Ad-Null- or Ad-Rgs5-transduced aoSMCs stimulated with ET-1 (100 nM, red: before stimulus, green: after stimulus, yellow: overlapping structures, scale bars: 10 µm).

**Figure 4 cells-12-00128-f004:**
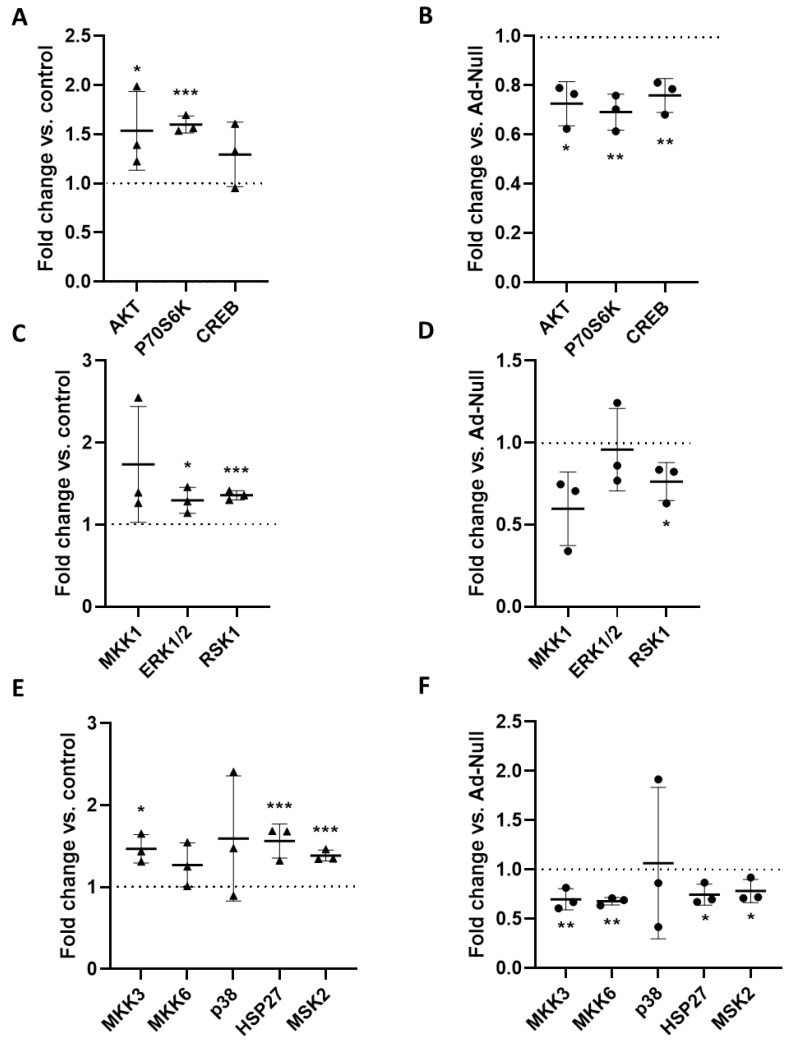
Analysis of kinase phosphorylation in aoSMC spheroids. Ad-Null- (**A**,**C**,**E**) or Ad-Rgs5 (**B**,**D**,**F**)-transduced murine aoSMC spheroids were stimulated for 15 min with either solvent (control) or ET-1 (100 nM). Protein lysates of the spheroids were subjected to a (phosphor)kinase protein array that detects the level of phosphorylation at the following (activating) phosphorylation sites of selected kinases: Akt (P-Ser473), P70S6K (P-Thr421/Ser424), CREB (P-Ser133), MKK1 (P-Ser217/Ser221), ERK1 (P-T202/Y204)/ERK2 (P-Y185/Y187), RSK1 (P-Ser380), MKK3 (P-Ser189), MKK6 (P-Ser207), p38 (P-Thr180/Tyr182), HSP27 (P-Ser82), and MSK2 (P-Ser360). The phosphorylation level was assessed densitometrically (**A**,**B**: AKT-related signaling pathway with P70S6K and CREB as AKT targets; **C**,**D**: MAP kinase kinase 1 (MKK1) → ERK1/2 → RSK1 signaling pathway; **E**,**F**: MAP kinase kinase 3/6 (MKK3/6) → p38 → HSP27/MSK2 signaling pathway, n = 3, * *p* < 0.05, ** *p* < 0.01, *** *p* < 0.001 vs. control or Ad-Null (dotted line, set to 1)). A representative array is shown as Appendix A.

**Figure 5 cells-12-00128-f005:**
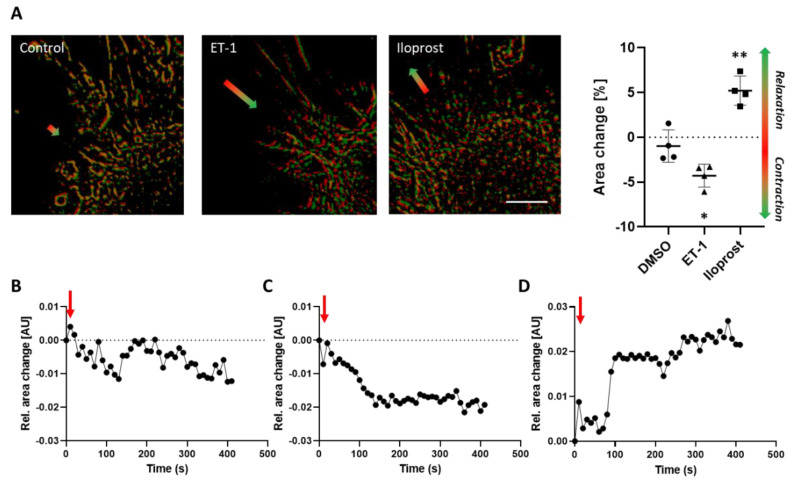
Contraction and relaxation of human umbilical artery smooth muscle cells (HUASMCs). HUASMC spheroids were attached to collagen type I hydrogels and stimulated with solvent (control), ET-1 (100 nM) or Iloprost (10 µM). Contraction/relaxation was recorded by time-lapse microscopy and quantified as mentioned before. Representative (enlarged) overlays of cellular segments indicate the direction of their movement ((**A**), colored arrows, scale bar: 500 µm), which were quantified accordingly ((**A**), n = 4, * *p* < 0.05, ** *p* < 0.01 vs. control). (**B**) Time-related structural changes (frame-based relative area changes) of selected spheroids after stimulation (red arrows, (**B**): DMSO, (**C**): ET-1, (**D**): Iloprost) were exemplarily assessed by computer-aided morphometric analyses.

## Data Availability

The full set of data as shown in Appendix A is available in the Gene Expression Omnibus (GEO) database: accession number: GSE133650.

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
