# Peer review of "Tracing G-Protein-Mediated Contraction and Relaxation in Vascular Smooth Muscle Cell Spheroids"

_cells, 2022, doi:10.3390/cells12010128_

Round 1
Reviewer 1 Report
The present study has focused on the development and characterization of an in vitro model of smooth muscle cells (SMCs) spheroids isolated from mice aorta. Given that a body of evidence has highlighted that cells cultured under standard 2D conditions lose many of their original features, the development of 3D models, among them spheroids, might provide more reliable and relevant results. In this context, the study showed an important concept, the 3d model presented herein is simple and allow further use for drug screening and toxicity assessment. The manuscript is interesting, well written, the information is clear and allow reproduction in other laboratories. There just a few points that I would recommend/suggest to further improvement of the manuscript.
1) Line 158 – it is written that cells were treated with "100 157 nM of ET-10 μM of YM-254890" , but I guess it should be ET or YM. Please check this information.
2) I recommend to show representative images of each target for the dot-blots (phosphokinase evaluation). Also, adding all images as Suppl Mat is recommended.
3) Please, explain which normalization test was used for the statistical analysis. There are data with 3 values per group, which does not allow application of normalization test, threfore is important to show how the authors proceeded.
4) On figure 3, graphs A and B show two bars, that I believe one is control while the other is treated with ET1, however it is not clear. I suggest to add different collours for the lines of the graphs and add this information in the legend for better understanding.
Author Response
The present study has focused on the development and characterization of an in vitro model of smooth muscle cells (SMCs) spheroids isolated from mice aorta. Given that a body of evidence has highlighted that cells cultured under standard 2D conditions lose many of their original features, the development of 3D models, among them spheroids, might provide more reliable and relevant results. In this context, the study showed an important concept, the 3d model presented herein is simple and allow further use for drug screening and toxicity assessment. The manuscript is interesting, well written, the information is clear and allow reproduction in other laboratories. There just a few points that I would recommend/suggest to further improvement of the manuscript.
1) Line 158 – it is written that cells were treated with "100 157 nM of ET-10 μM of YM-254890" , but I guess it should be ET or YM. Please check this information.
We thank the reviewer for these constructive comments and for thoroughly reading the manuscript. We corrected this typo. It should read “100 nM of ET-1 or 10 μM of YM-254890”
2) I recommend to show representative images of each target for the dot-blots (phosphokinase evaluation). Also, adding all images as Suppl Mat is recommended.
We added representative images of a complete dot blot as Supplementary Figure S5.
3) Please, explain which normalization test was used for the statistical analysis. There are data with 3 values per group, which does not allow application of normalization test, therefore is important to show how the authors proceeded.
We agree that most statistical normalization tests such as the Kolmogorov-Smirnov test are not compatible with low sample numbers. We therefore applied the Shapiro-Wilk test as suggested for small sample sizes, which are dependent on one variable. For instance, the following values were determined for the values as shown in Figure 4A:
|
Shapiro-Wilk test |
AKT |
P70S6K |
CREB |
|
W |
0.9047 |
0.8667 |
0.9930 |
|
P value |
0.4005 |
0.2863 |
0.8404 |
|
Passed normality test (alpha=0.05)? |
Yes |
Yes |
Yes |
|
P value summary |
ns |
ns |
ns |
We now mention the test in the Material & Methods section.
4) On figure 3, graphs A and B show two bars, that I believe one is control while the other is treated with ET1, however it is not clear. I suggest to add different collours for the lines of the graphs and add this information in the legend for better understanding.
We thank for this suggestion and have now used colored symbols to distinguish between control and ET-1 treatment as suggested.
Reviewer 2 Report
The paper describes a novel method for detecting smooth muscle cell function and researching G protein-mediated vasoconstriction/vasorelaxation.
Only two concerns:
1. How precise is video-based spheroid contraction? Is the morphology change close to mimicking the function of the VSMC?
2. How consistent is the system in detecting small spheroid differences?
Author Response
The paper describes a novel method for detecting smooth muscle cell function and researching G protein-mediated vasoconstriction/vasorelaxation.
Only two concerns:
- How precise is video-based spheroid contraction? Is the morphology change close to mimicking the function of the VSMC?
We thank the reviewer for these comments. The assay was internally evaluated by two research assistants, which were able to reproduce the results of each other. Most adhered smooth muscle cells contract along their longitudinal axis, mimicking their behavior in vivo. We also observe that VSMCs located in the spheroids contribute to the overall relaxation/contraction response. The morphological changes of these cells, however, were not traceable for optical reasons. Collectively, the VSMCs appear to mimic some relevant aspects of VSMC functions limited by the artificial environment.
- How consistent is the system in detecting small spheroid differences?
By applying simple analytical techniques including the whole spheroid and considering the standard deviation, differences larger than 30 % are detectable. However, by restricting the quantitative analyses on changes observed along the main contraction/relaxation vector, even smaller differences can be assessed. Moreover, instead of determining general area changes, the moving distance of individual cellular elements may be directly assessed to further increase the resolution of this assay.